# Regional and temporal variations in COVID-19 cases and deaths in Ethiopia: Lessons learned from the COVID-19 enhanced surveillance and response

Gizaw Teka[1], Adane Woldeab[1,2], Nebiyu Dereje[3]*, Frehywot Eshetu[4],
Lehageru Gizachew[2], Zelalem Tazu[2], Leuel Lisanwork[2], Eyasu Tigabu[2],
Ayele Gebeyehu[1], Adamu Tayachew[1], Mengistu Biru[1], Tsegaye Berkessa[1],
Abrham Keraleme[1], Fentahun Bikale[1], Wolde Shure[1], Admikew Agune[1],
Bizuwork Haile[1], Beza Addis[1], Muluken Moges[1], Melaku Gonta[1], Aster Hailemariam[1],
Laura Binkley[5], Saira Nawaz[5], Shu-Hua Wang[5,6], Zelalem Mekuria[5], Ayalew Aklilu[3],
Jemal Aliy[3], Sileshi Lulseged[3], Abiy Girmay[7], Abok Patrick[7], Berhanu Amare[4],
Hulemenaw Delelegn[4], Sharon Daves[4], Getnet Yimer[8], Ebba Abate[2], Mesfin Wossen[1‡],
Zenebe Melaku[3‡], Wondwossen Gebreyes[5,9‡], Desmond E. Williams[4‡],
Aschalew Abayneh[1‡]

1 Ethiopian Public Health Institute, Addis Ababa, Ethiopia, 2 The Ohio State University Global One Health initiative (GOHi), Addis Ababa, Ethiopia, 3 ICAP at Columbia University, Addis Ababa, Ethiopia, 4 Division of Global Health Protection (DGHP), US Centers for Disease Control and Prevention (CDC), Addis Ababa, Ethiopia, 5 The Ohio State University Global One Health initiative (GOHi), Columbus, Ohio, United States of America, 6 Department of Internal Medicine, The Ohio State University, Columbus, Ohio, United States of America, 7 World Health Organization (WHO) Ethiopia Office, Addis Ababa, Ethiopia, 8 University of Pennsylvania, Perelman School of Medicine, Pennsylvania, Philadelphia, United States of America, 9 Department of Veterinary Preventive Medicine, The Ohio State University, Columbus, Ohio, United States of America

☉ These authors contributed equally to this work.
‡ These authors also contributed equally to this work.
* neba.jahovy@gmail.com

## Abstract

### Background

The COVID-19 pandemic is one of the most devastating public health emergencies of international concern to have occurred in the past century. To ensure a safe, scalable, and sustainable response, it is imperative to understand the burden of disease, epidemiological trends, and responses to activities that have already been implemented. We aimed to analyze how COVID-19 tests, cases, and deaths varied by time and region in the general population and healthcare workers (HCWs) in Ethiopia.

### Methods

COVID-19 data were captured between October 01, 2021, and September 30, 2022, in 64 systematically selected health facilities throughout Ethiopia. The number of health facilities included in the study was proportionally allocated to the regional states of Ethiopia. Data

**Data Availability Statement:** All data underlying the findings described in the manuscript are provided as a supplementary file (S1 Data).

**Funding:** This study was conducted through the analysis of COVID-19 surveillance and response data captured from health facilities included in the U.S. CDC-funded CARES Act project. Its contents are solely the responsibility of the authors and do not necessarily represent the official views of the US Centers for Disease Control and Prevention (CDC) or the Department of Health and Human Services.

**Competing interests:** The authors have declared that no competing interests exist.

were captured by standardized tools and formats. Analysis of COVID-19 testing performed, cases detected, and deaths registered by region and time was carried out.

## Results

We analyzed 215,024 individuals' data that were captured through COVID-19 surveillance in Ethiopia. Of the 215,024 total tests, 18,964 COVID-19 cases (8.8%, 95% CI: 8.7%–9.0%) were identified and 534 (2.8%, 95% CI: 2.6%– 3.1%) were deceased. The positivity rate ranged from 1% in the Afar region to 15% in the Sidama region. Eight (1.2%, 95% CI: 0.4%– 2.0%) HCWs died out of 664 infected HCWs, of which 81.5% were from Addis Ababa. Three waves of outbreaks were detected during the analysis period, with the highest positivity rate of 35% during the Omicron period and the highest rate of ICU beds and mechanical ventilators (38%) occupied by COVID-19 patients during the Delta period.

## Conclusions

The temporal and regional variations in COVID-19 cases and deaths in Ethiopia underscore the need for concerted efforts to address the disparities in the COVID-19 surveillance and response system. These lessons should be critically considered during the integration of the COVID-19 surveillance system into the routine surveillance system.

## Introduction

The COVID-19 pandemic has been one of the most devastating Public Health Emergencies of International Concern (PHEIC) of this century, with a high burden of cases and deaths globally [1]. The COVID-19 pandemic has tested the resilience of the global public health system and threatened the existing fragile public health systems [2,3]. Several global and local initiatives and preventive measures were implemented to contain the spread of the disease in Ethiopia. Some of these measures were a consequence of the World Health Organization's (WHO) decision to declare COVID-19 as a PHEIC [4] and of the Ethiopian government to declare a State of Emergency, where travel restrictions, school closures, and a ban on public gatherings were implemented. Moreover, with the rapid advent and introduction of the COVID-19 vaccines, the response efforts of the country have been improved [5]. However, vaccine hesitancy was one of the pressing challenges [6].

The WHO recommended COVID-19 surveillance as one of the critical measures necessary to end the pandemic and effectively inform public health responses to limit the spread of the disease and its impacts [7]. New cases and clusters of infections must be rapidly identified before widespread transmission occurs [8]. Surveillance for COVID-19 is important to understand long-term epidemiological trends of morbidity and mortality, early warning of changes in epidemiological patterns, the burden of disease on healthcare capacity, circulation of known variants of concern (VOCs), and early detection of new variants of concern. Thus, it was critical to build a resilient public health surveillance system to allow early detection, prevention, and response of COVID-19 in an effective and timely manner [7–9]. To this end, Ethiopia implemented several COVID-19 surveillance and response activities to reduce the spread of the pandemic which included active case detection, contact tracing, care and isolation, and quarantine [10–12].

To strengthen surveillance for COVID-19 and other reportable diseases, a Coronavirus Relief and Economic Security (CARES) Act was enacted by the U.S. government. The U.S.

Centers for Disease Control and Prevention (CDC) funded the Ethiopian National Surveillance Project with CARES funding through a cooperative agreement [13]. The project allowed the implementation of different activities through partnerships among the Ethiopian Public Health Institute (EPHI), Regional Health Bureaus (RHBs), The Ohio State University Global One Health initiative (GOHi), the International Center for AIDS Program (ICAP) at Columbia University and other partners. Through the project, key interventions to augment the existing COVID-19 surveillance and response capacity were implemented at both national and regional levels in selected healthcare facilities throughout Ethiopia. Accordingly, COVID-19 surveillance and response data were actively captured from supported healthcare facilities and reported to the EPHI in a systematic and timely manner. As COVID-19 emergency phase ends, Ethiopia is working to integrate its surveillance into the routine public health surveillance system as per the WHO recommendations [14]. Consequently, it is critical to adequately understand the burden of the disease, patterns of outbreaks, and response activities.

Only a few regional studies on the clinical and epidemiological profile of COVID-19 cases in Ethiopia have been published [15–17]. These studies reported a limited number of cases and lacked information related to the epidemiological and temporal link among the different waves of outbreaks. In addition, prior studies do not adequately correlate surveillance and response efforts carried out during the outbreaks, nor did the studies analyze the burden of the disease among HCWs. To ensure a safe, scalable, and sustainable transition of the surveillance system, it is imperative to understand trends and responses to COVID-19 from the analysis of data captured through this collaborative project. Here we describe the burden of COVID-19 (morbidity and mortality), epidemiological trends, the burden on the healthcare capacity, and response to control spread during the project implementation period.

## Materials and methods

### Study setting, design, and selection of healthcare facilities

The public health surveillance system in Ethiopia follows the healthcare delivery systems structure with administrative hierarchies within the national health system including the primary healthcare level (health posts, health centers, districts (a local name "woreda"), hospitals), secondary healthcare level (general hospitals), and tertiary healthcare level (specialized comprehensive hospitals). The woreda, zonal, regional, and federal (EPHI) health offices are responsible for coordinating the overall public health surveillance and response systems for each respective level [18]. COVID-19 surveillance was structured similarly, utilizing these healthcare levels [19].

The healthcare facilities (health centers and hospitals) were responsible for surveilling COVID-19 cases and daily reporting to higher administrative levels. Our descriptive, cross-sectional study was conducted based on the COVID-19 surveillance and response data captured from 64 of these selected healthcare facilities between October 01, 2021 and September 30, 2022. Among the selected healthcare facilities, 31 were primary level, 21 were secondary level, and 12 were tertiary level healthcare facilities. The selection of facilities was systematically based on reported patient load (high-load facilities were selected), availability of COVID-19 testing service, and access to electricity. The number of healthcare facilities included was proportionally allocated to the regional states of Ethiopia to ensure representation of all geographical areas of the country. However, due to the conflict in Northern Ethiopia (Tigray region), we were unable to include that region in this study. The selected health facilities were supported by the CARES Act project through the provision of designated public health surveillance officers responsible for carrying out COVID-19 surveillance activities to capture data and share reports to the next level in the health system.

## Data collection tools and procedures

To ensure the quality of surveillance data and minimize variability, a team of experts and researchers from the collaborating entities (EPHI, ICAP and GOHi) provided standardized training on COVID-19 surveillance to the public health surveillance officers. The team has also provided continuous mentorship and supportive supervision of officers at reporting facilities regularly (every two months). The officers at regional and central levels generated surveillance data completeness and timeliness reports and provided feedback continuously to the reporting facilities.

Surveillance data were captured in healthcare facilities using standardized COVID-19 surveillance tools and case definitions (S1 Appendix) adapted from the national COVID-19 surveillance guideline and the WHO COVID-19 surveillance interim guideline [10,20]. The deployed public health surveillance officers (Master of Public Health degree holders) were responsible for capturing COVID-19 surveillance data from clients, registering cases in the COVID-19 surveillance logbooks, and reporting to higher levels through the District Health Information System (DHIS2) or via e-mailing of Excel spreadsheets. The COVID-19 surveillance tools captured data on suspect case identification and testing status, number of confirmed cases, COVID-19 testing modality, number of admissions and deaths, contacts identified and tested, number of confirmed HCW cases and deaths, number of clusters/outbreaks investigated, and responses provided. Data regarding the circulating variants of concern was obtained from the EPHI COVID-19 testing laboratory [10] and the World Health Organization's SARS-COV-2 tracking system [21]. Cluster investigation was conducted following WHO standard procedures (S1 Appendix) [22].

## Data management and analysis

Public health surveillance data were retrieved from different sources (e.g., DHIS2 and Excel spreadsheets) and compiled into one master Excel sheet. We then cleaned and checked the data for completeness, and exported it to SPSS version 25 software for analysis. Descriptive analysis (frequency and proportions) was carried out and presented in tables and figures while trends of COVID-19 tests, cases, and deaths were presented in line graphs. We analysed the case fatality rate (CFR), positivity rate, and hospitalization rates and their respective 95% confidence intervals (CI). We used R software to create the maps using the base map shape file from https://data.humdata.org/dataset/cod-ab-eth. Calculations for all metrics can be found in Table 1. For analysis, all data were stratified by regional state.

## Ethical considerations

This study utilized routine COVID-19 surveillance data; no identifiable, patient-level information was used. The routine COVID-19 surveillance was conducted based on the nationally approved protocol and we analysed secondary data collected for surveillance purpose.

## Role of funding source

The funder had no role in the design, processing, analysis, and interpretation of this study.

## Results

This study analyzed data of 215,024 COVID-19 suspected cases, contacts, and travelers, who were surveilled in 64 health facilities in Ethiopia. The average completeness and timeliness of the data were 98.1% and 97.9%, respectively (Fig 1).

**Table 1. Facility-level COVID-19 surveillance metrics and respective calculations used.**

| Metric | Calculation |
|---|---|
| COVID-19 Positivity Rate | Number of confirmed cases/total number of tests |
| Case Fatality Rate | Number of confirmed COVID-19-induced deaths/total number of confirmed COVID-19 cases |
| Hospitalization Rate | Total number of patients hospitalized for COVID-19/total number of confirmed COVID-19 cases |
| Intensive Care Unit (ICU) Admission Rate | Total number of cases admitted to the ICU/total number of hospitalized patients |
| Rate of mechanical ventilators occupied | Total number of mechanical ventilators used / the total number of ventilators in each hospital |
| Rate of bed occupied | Total number of beds in a facility occupied by COVID-19 patients/ total number of beds in the facility |
| Surveillance data completeness | Average of the number of health facilities that reported in the Epi-week/total number of health facilities expected to report |
| Surveillance data timeliness | Average of the number of health facilities that reported to EPHI within the prescribed time/total number of health facilities expected to report within the prescribed time (every Tuesday) |

## COVID-19 testing and case detection

Out of the total COVID-19 suspected cases, contacts, and travelers tested, 88.9% (95% CI: 88.8% - 89.1%) were tested by Antigen Detection Rapid Diagnostic Test (Ag RDT) and 11.1% (95% CI: 10.9% - 11.2%) were tested by Real-Time Polymerase Chain Reaction (RT-PCR). However, the Harari and Benishangul Gumuz region facilities used only Ag RDT (100%) as a testing modality. The majority of tests were performed at the secondary healthcare level (general hospitals). One-third (33%) of the testing and one-third (33%) of cases were from facilities in Addis Ababa (Fig 2). Among those tested, 18,964 were found to be positive for COVID-19, with a positivity rate of 8.8% (95% CI: 8.7%– 9.0%). Of the positive cases, (85.4%, 95%CI:

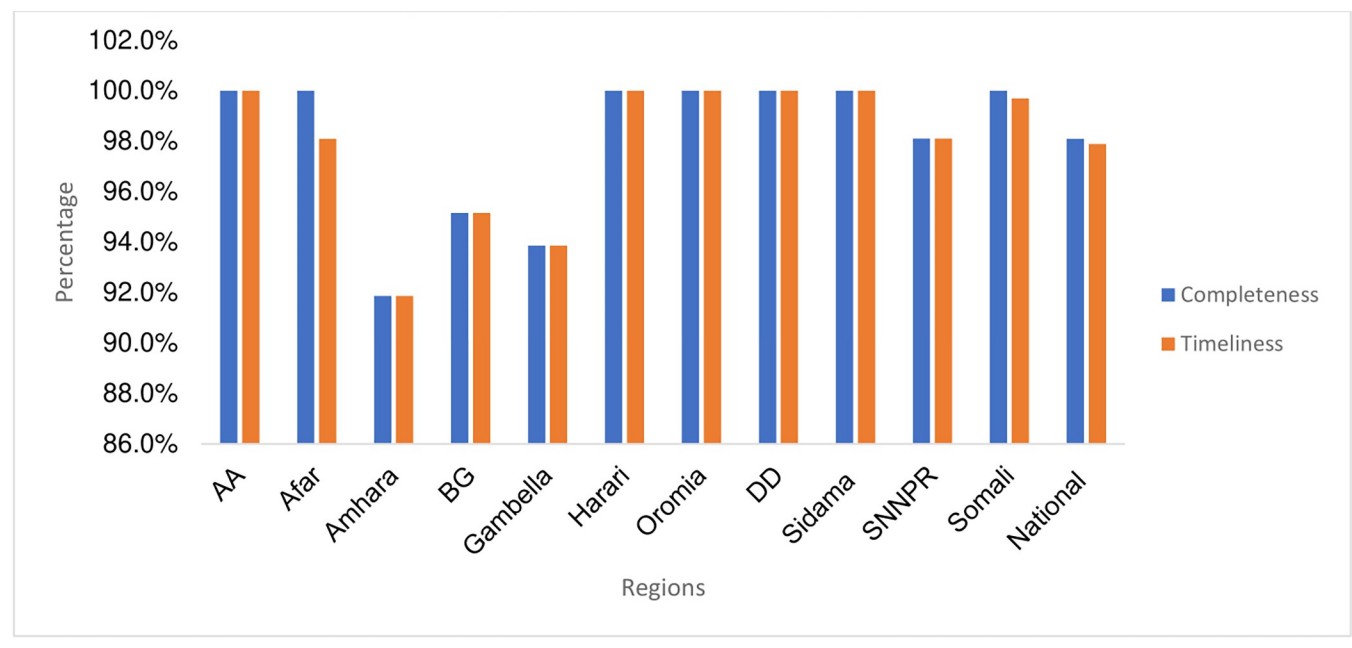

**Fig 1. Completeness and timeliness of the COVID-19 surveillance data by region.**

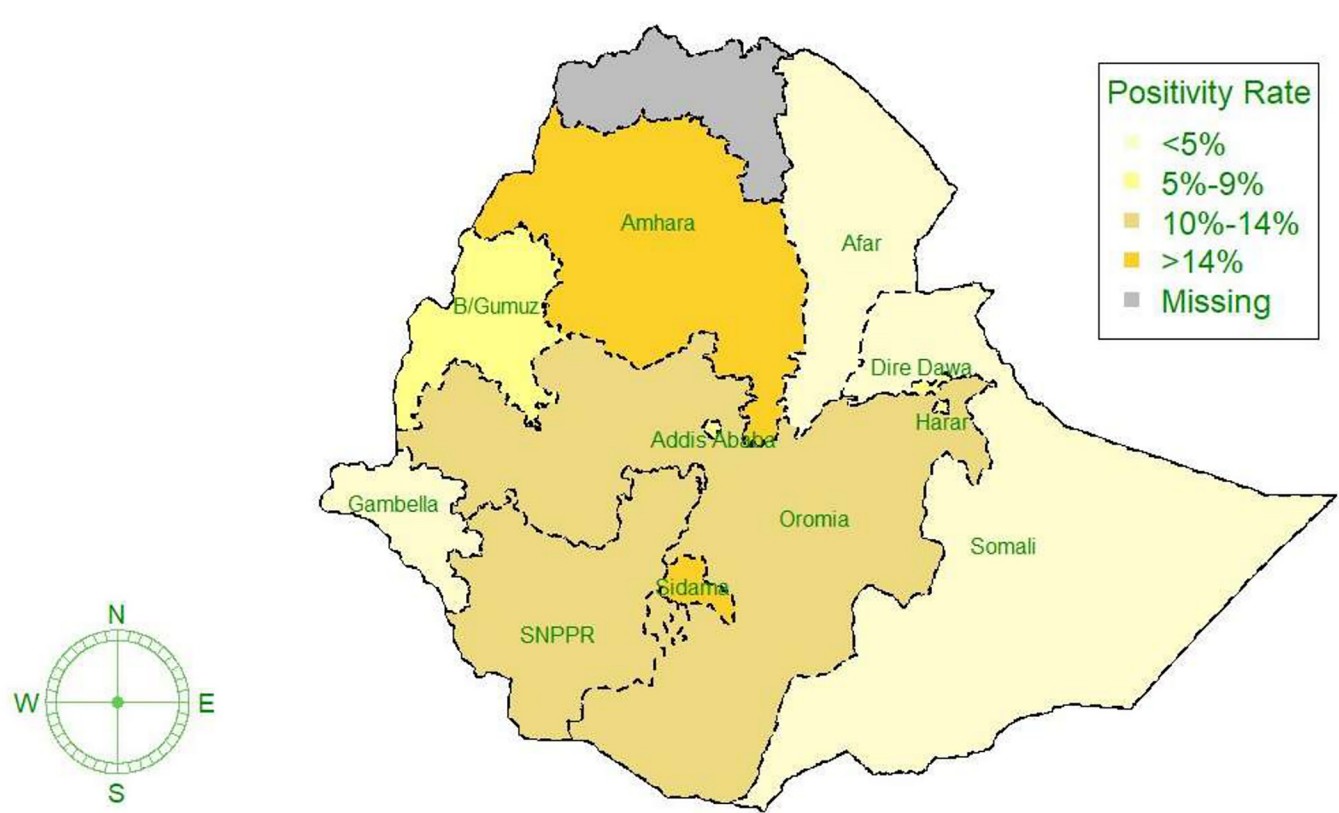

**Fig 2. Regional distribution of COVID-19 cases in Ethiopia from Oct 01, 2021 –Sept 30, 2022 (source of the base map shape file: https://data.humdata. org/dataset/cod-ab-eth).**

85.1% - 85.9%) were tested by Ag-RDT. The positivity rate ranged from 1% in the Afar region to 15% in the Sidama region (Table 2).

Three waves of outbreaks were detected during the analysis period. The first peak occurred during Epi-weeks 39 and 40 of 2021 (October 2021). The second peak occurred from Epi-week 50 of 2021 to Epi-week 2 of 2022 (Dec 12, 2021 –Jan 15, 2022) and sharply declined from Epi-week 2 to Epi-week 4 of 2022 (Jan 15–31, 2022). During this second peak period, the positivity rate reached as high as 35%. The curve remained flat until Epi-week 20 of 2022 (May 13, 2022) after which it began to rise and a third wave was detected from Epi-week 24 to Epi-week 26 of 2022 (June 12 –July 02, 2022) after which the curve declined (Fig 3).

### COVID-19 related hospitalization and deaths

Among the confirmed COVID-19 cases, 2573 (13.6%, 95%CI: 13.1% - 14.1%) patients were hospitalized. More than two-thirds (67%) of the hospitalizations were recorded during Epi-week 16 of 2022 (April 16, 2022). However, the highest rate of ICU beds and mechanical ventilators (38%) were occupied by COVID-19 patients during Epi-week 40 of 2021 (October 2021) (Fig 4). One out of five (21%, 95% CI: 19.4% - 22.6%) patients hospitalized for COVID-19 (2,573 patients) died (534 patients), with more than a quarter (26.3%) of the deaths recorded at primary Health Care Units (HCUs) (Table 3).

There were 534 deaths reported among the confirmed cases (18,964) with a total case fatality rate (CFR) of 2.8% (95% CI: 2.6%– 3.1%). The highest CFR (7.4%) was recorded in the

**Table 2. Tests, confirmed cases, and positivity rate by regions of Ethiopia.**

| Regions | Total tests | Tests by testing modality | | Test by healthcare facility type | | | Confirmed case (Positivity rate) | Confirmed case (Positivity rate) by facility type | | | Confirmed case (Positivity rate) by testing modality | |
|---|---|---|---|---|---|---|---|---|---|---|---|---|
| | | Ag-RDT | RT-PCR | Primary level | Secondary level | Tertiary level | | Primary level | Secondary level | Tertiary level | Ag-RDT | RT-PCR |
| Addis Ababa | 71094 | 49684 (69.9%) | 21410 (30.1%) | 5764 (8.1%) | 32132 (45.2%) | 33198 (46.7%) | 6212 (8.7%, 95%CI: 8.5%-8.9%) | 1174 (18.9%) | 3455 (55.6%) | 1583 (25.5%) | 4286 (69.0%) | 1926 (31.0%) |
| Afar | 8036 | 8026 (99.9%) | 10 (0.1%) | 7609 (94.7%) | 427 (5.3%) | 0(0%) | 77 (1%, 95%CI: 0.8%–1.2%) | 68 (88.3%) | 9 (11.7%) | 0 (0%) | 75 (97.4%) | 2 (2.6%) |
| Amhara | 10535 | 10520 (99.9%) | 10 (0.1%) | 2713 (25.8%) | 675 (6.4%) | 7147 (67.8%) | 1531 (14.6%, 95% CI: 13.9%-15.2%) | 326 (21.3%) | 37 (2.4%) | 1168 (76.3%) | 1526 (99.7%) | 5 (0.3%) |
| Benishangul Gumuz | 9019 | 9019 (100.0%) | 0 (0%) | 5489 (60.9%) | 3530 (39.1%) | 0(0%) | 595 (6.6%, 95% CI: 6.1–7.1) | 252 (42.4%) | 343 (57.6%) | 0 (0%) | 595 (100%) | 0 |
| Dire Dawa | 31603 | 31587 (99.9%) | 10 (0.1%) | 15214 (48.1%) | 16389 (51.9%) | 0(0%) | 2293 (7.3%, 95% CI: 7.0–7.6) | 1186 (51.7%) | 1107 (48.3%) | 0 (0%) | 2290 (99.9%) | 3 (0.1%) |
| Gambella | 6706 | 6687 (99.7%) | 19 (0.3%) | 6261 (93.4%) | 445 (6.6%) | 0(0%) | 210 (3.1%, 95% CI: 2.7%-3.6%) | 176 (83.8%) | 34 (16.2%) | 0 (0%) | 201 (95.7%) | 9 (4.3%) |
| Harari | 13093 | 13093 (100%) | 0 (0%) | 5951 (45.5%) | 3580 (27.3%) | 3562 (27.2%) | 1181 (9.0%, 95% CI: 8.5%-9.5%) | 375 (31.8%) | 640 (54.2%) | 166 (14.1%) | 1181 (100%) | 0 |
| Oromia | 34446 | 34129 (99.1%) | 317 (0.9%) | 18616 (54.0%) | 15830 (46.0%) | 0(0%) | 4076 (11.8%, 95% CI: 11.5–12.2) | 2584 (63.4%) | 1492 (36.6%) | 0 (0%) | 3872 (95.0%) | 204 (5.0%) |
| Sidama | 6892 | 5316 (77.1%) | 1576 (22.9%) | 0(0%) | 5307 (77.0%) | 1585 (23.0%) | 1077 (15.6%,95% CI: 14.8%-16.5%) | 0 (0%) | 862 (80.0%) | 215 (20.0%) | 612 (56.8%) | 465 (43.2%) |
| SNNPR | 13549 | 13410 (99.0%) | 139 (1.0%) | 0(0%) | 4285 (31.6%) | 9264 (68.4%) | 1305 (9.6%, 95% CI: 9.1%-10.1%) | 0 (0%) | 489 (37.5%) | 816 (62.5%) | 1240 (95.0%) | 65 (5.0%) |
| Somali | 10051 | 9741 (96.9%) | 310 (3.1%) | 3583 (35.6%) | 5393 (53.7%) | 1075 (10.7%) | 407 (4%, 95%CI: 3.7%-4.4%) | 129 (31.7%) | 273 (67.1%) | 0 (0%) | 320 (68.6%) | 87 (31.4%) |
| Grand Total | 215024 | 191212 (88.9%, 95%CI: 88.8%-89.1%) | 23812 (11.1%, 95% CI: 10.9% -11.2%) | 71200 (33.1%, 95% CI: 32.9% -33.3%) | 87993 (40.9%, 95% CI: 40.7%-41.1%) | 55831 (26.0%, 95%CI: 25.8%-26.1%) | 18964 (8.8%, 95% CI: 8.7%–9.0%) | 6270 (33.1%, 95%CI: 32.4% -33.7%) | 8741 (46.1%, 95% CI: 45.4% -46.8%)) | 3953 (20.8%, 95%CI: 20.3% -21.4%) | 16,198 (85.4%, 95%CI: 85.1% -85.9%) | 2,766 (14.6%, 95%CI: 14.1% -14.9%) |

Somali region and the highest total number of deaths was reported in Addis Ababa (Fig 5, Table 3).

## COVID-19 Healthcare Workers (HCWs) infection and death

Six hundred sixty-four (3.5%) of the confirmed COVID-19 cases were HCWs, eight of whom died due to COVID-19 (1.2%, 95% CI: 0.37%– 2.03%). Of the infected HCWs, 81.5% were

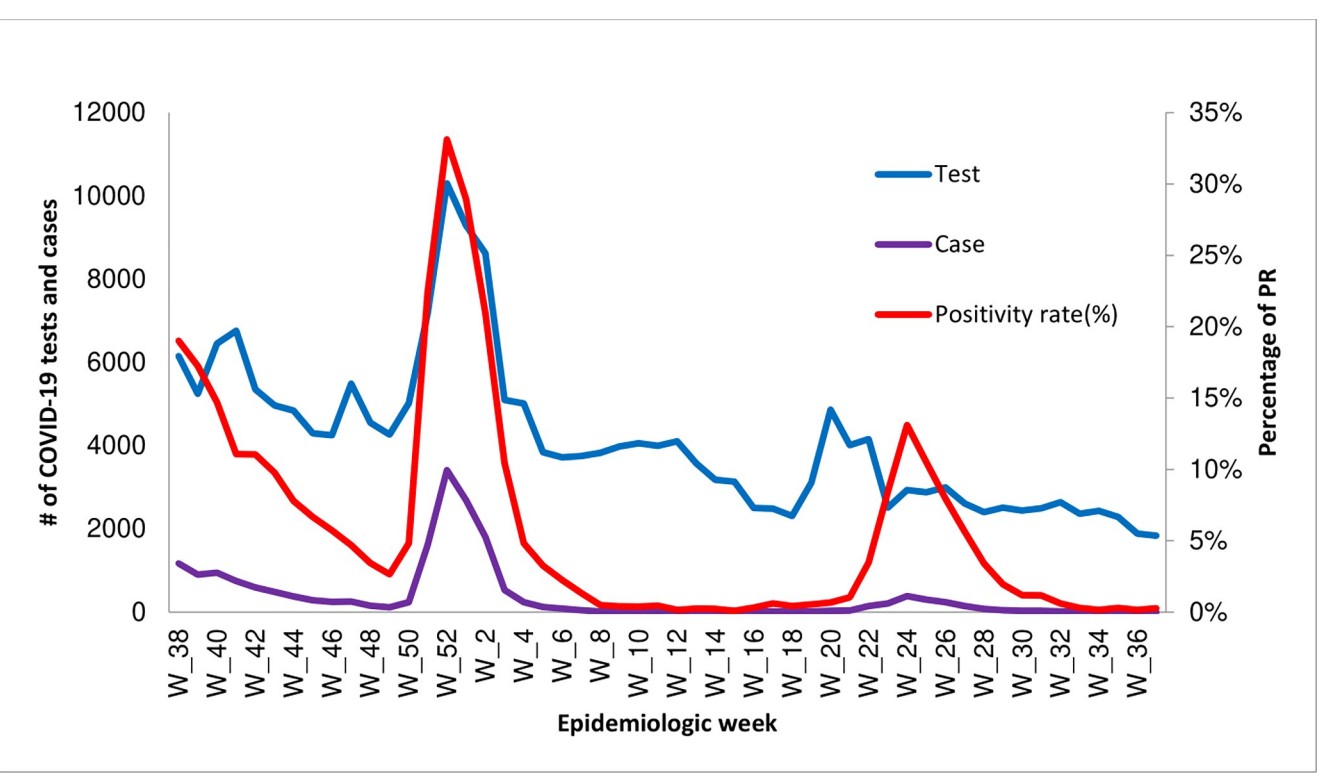

**Fig 3. The trend of COVID-19 tests, cases, and positivity rate in Ethiopia from Oct 01, 2021 Sept 30, 2022.**

from Addis Ababa, and nearly two-thirds (61.6%) were from general hospitals (secondary level healthcare unit) (Table 4). All the recorded deaths of HCWs were from Addis Ababa.

## COVID-19 response

A total of 12,813 contacts were identified through contact tracing measures of which 9,323 (72.8%) were tested and 1,095 (11.8%, 95%CI: 11.1% - 12.4%) were found to be positive for COVID-19. More than 200 clusters of outbreaks were identified and investigated across all regions of Ethiopia, with the greatest number of clusters being investigated in the Benishangul Gumuz Region (Table 5).

## Discussion

The principal findings of this study indicate the COVID-19 case and death tolls, positivity rate, and CFR were substantial among the study population as a whole and high among HCWs. COVID-19 has had a tremendous impact on the overall health, social, psychological, and economic well-being of the Ethiopian population [2,11,23,24]. The resiliency and capacity of the public health system were also highly threatened. Moreover, the findings revealed regional variations in COVID-19 testing, testing algorithms, case detection, hospitalization, contact tracing, and cluster investigations. It is important to note the regional disparities in COVID-19 surveillance and response activities could pose challenges to the efforts to control the pandemic and prepare for future public health emergencies. Sustainable and effective integration of the surveillance system into the routine public health surveillance system should consider addressing these regional variations by properly assessing and identifying the gaps. Notably,

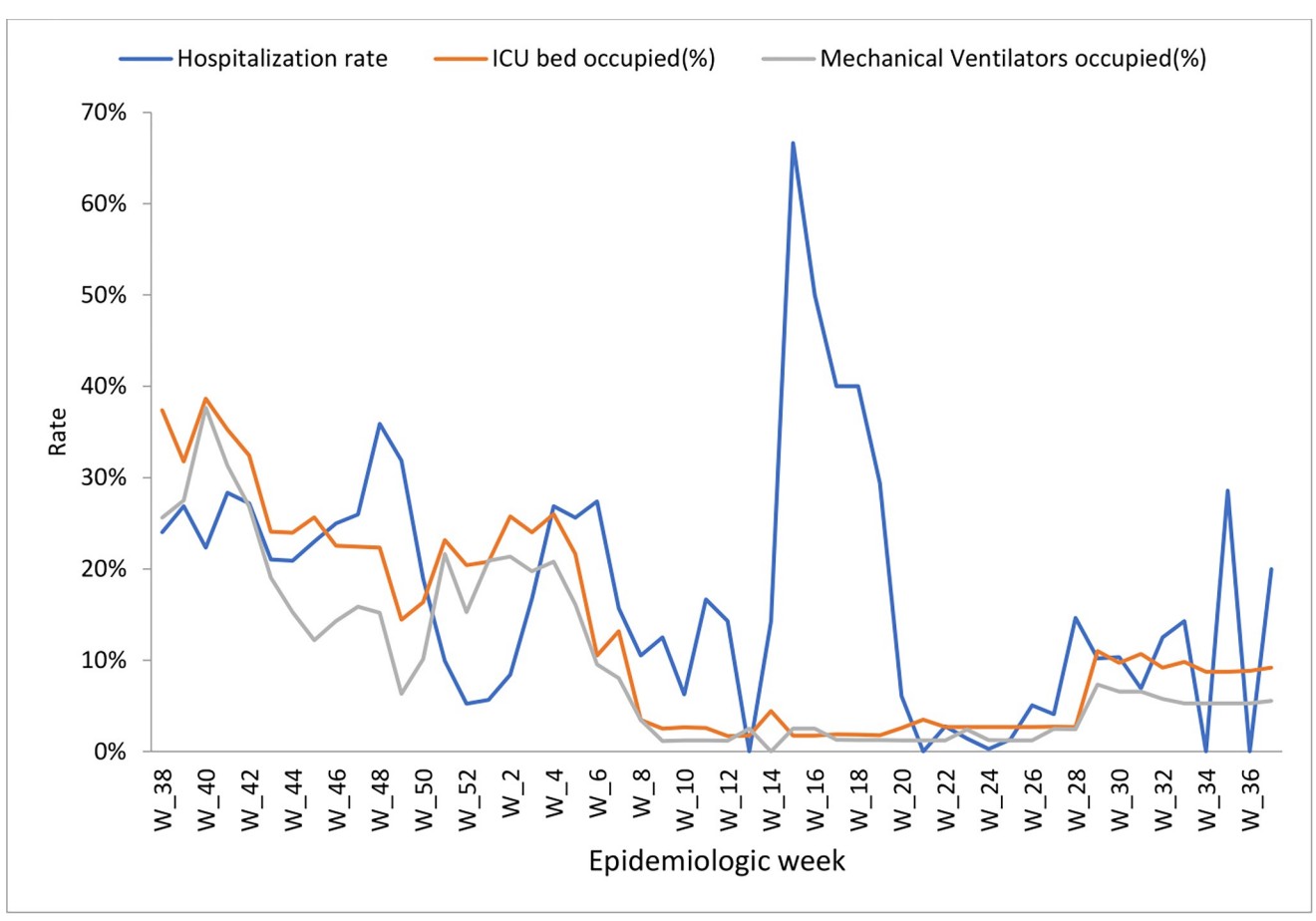

**Fig 4. COVID-19 hospitalization, ICU beds, and mechanical ventilators occupied in Ethiopia (Oct 01, 2021 –Sept 30, 2022).**

**Table 3. COVID-19 hospitalization and deaths by region and facility type in Ethiopia.**

| Region | Confirmed cases | Death | CFR (%) | Hospitalized cases by facility type (n, %) | | | Death by facility type (n, %) | | |
|---|---|---|---|---|---|---|---|---|---|
| | | | | Primary HCU | Secondary HCU | Tertiary HCU | Primary HCU | Secondary HCU | Tertiary HCU |
| Addis Ababa | 6212 | 135 | 2.2(95%CI: 1.8–2.2) | 0(0) | 401(56.9) | 304(43.1) | 0(0) | 43(10.7) | 92(30.3) |
| Afar | 77 | 2 | 2.6(95%CI: 0.3–9.0) | 4(44.4) | 5(55.6) | 0(0) | 1(25.0) | 1(20.0) | 0(0) |
| Amhara | 1531 | 85 | 5.6(95%CI:4.5–6.8) | 9(2.1) | 16(3.8) | 397(94.1) | 0(0) | 4(25.0) | 81(20.4) |
| Benishangul | 595 | 0 | 0.00 | 0(0) | 8(100) | 0(0) | 0(0) | 0(0) | 0(0) |
| Dire Dawa | 2293 | 31 | 1.4(95%CI: 0.9–1.9) | 0(0) | 234(100) | 0(0) | 0(0) | 31(13.2) | 0(0) |
| Gambella | 210 | 0 | 0.00 | 1(100.0) | 0(0) | 0(0) | 0(0) | 0(0) | 0(0) |
| Harari | 1181 | 33 | 2.8(95%CI: 1.9–3.9) | 2(1.7) | 7(5.8) | 112(92.6) | 0(0) | 0(0) | 33(29.5) |
| Oromia | 4076 | 123 | 3.0(95%CI: 2.5–3.6) | 121(27.8) | 314(72.2) | 0(0) | 33(27.3) | 90(28.7) | 0(0) |
| Sidama | 1077 | 40 | 3.7(95%CI: 2.7–5.0) | 0(0) | 80(34.8) | 150(65.2) | 0(0) | 0(0) | 40(26.7) |
| SNNPR | 1305 | 55 | 4.2(95%CI: 3.2–5.5) | 0(0) | 56(18.3) | 250(81.7) | 0(0) | 13(23.2) | 42(16.8) |
| Somali | 407 | 30 | 7.4(95%CI: 5.0–10.4) | 15(14.7) | 86(84.3) | 1(1.0) | 6(40.0) | 23(26.7) | 1(100) |
| Grand Total | 18964 | 534 | 2.8(95%CI:2.6–3.1) | 152(5.9) | 1207(46.9) | 1214(47.2) | 40(26.3) | 205(17.0) | 289(23.8) |

CFR–Case fatality rate; HCU–Healthcare unit.

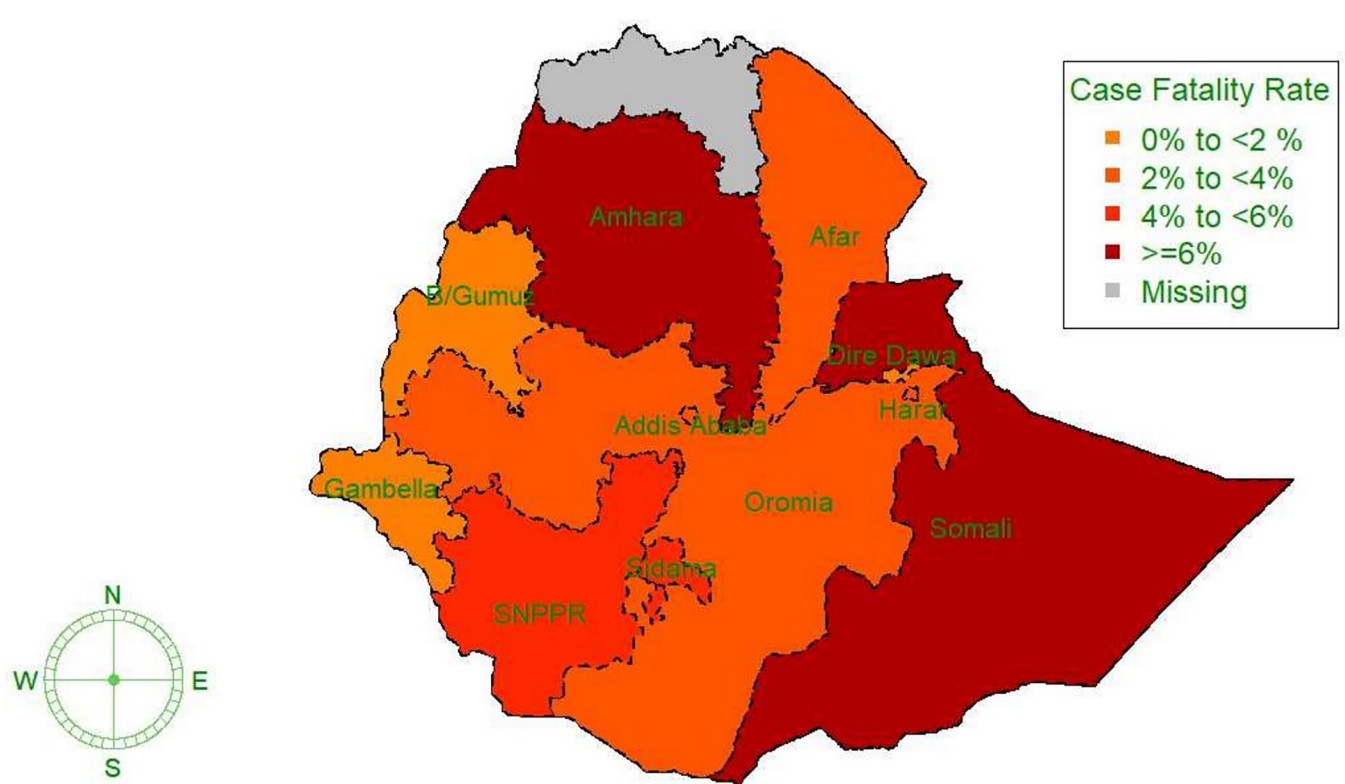

**Fig 5. Number of COVID-19 deaths in regions of Ethiopia from October 01, 2021 –September 30, 2022 (source of the base map shape file: https://data. humdata.org/dataset/cod-ab-eth).**

improving coordination and cooperation among surveillance and response systems in different regions to ensure more equitable and consistent response measures are critical.

The completeness and timeliness of the COVID-19 surveillance data in our study surpassed the national and WHO standards [7,10]. This was achieved by the implementation of several

**Table 4. Healthcare workers' infection and death by region and facility type.**

| Region | HCW confirmed cases by facility type | | | | HCW death by facility type (Fatality rate) | | | |
|---|---|---|---|---|---|---|---|---|
| | Primary healthcare unit | Secondary healthcare unit | Tertiary healthcare unit | Total | Primary healthcare unit | Secondary healthcare unit | Tertiary healthcare unit | Total |
| Addis Ababa | 6(1.5%) | 318(81.5%) | 66(16.9%) | 390 | 1(12.5%) | 3(37.5%) | 4(50.0%) | 8 (100%) |
| Afar | 3(100%) | 0(0%) | 0(0%) | 3 | | | | |
| Amhara | 5(10.0%) | 1(2.0%) | 44(88.0%) | 50 | | | | |
| Benishangul | 8(100%) | 0(0%) | 0(0%) | 8 | | | | |
| Dire Dawa | 14(32.6%) | 29(67.4%) | 0(0%) | 43 | | | | |
| Gambella | 1(100%) | 0(0%) | 0(0%) | 1 | | | | |
| Harari | 4(12.1%) | 6(18.2%) | 23(67.7%) | 33 | | | | |
| Oromia | 42(53.8%) | 36(46.2%) | 0(0%) | 78 | | | | |
| Sidama | 0(0%) | 7(33.3%) | 14(66.7%) | 21 | | | | |
| SNNPR | 0(0%) | 2(9.1%) | 20(90.9%) | 22 | | | | |
| Somali | 5(33.3%) | 10(66.7%) | 0(0%) | 15 | | | | |
| Grand Total | 88(13.3%, 95%CI: 10.8%-16.1%) | 409(61.6%, 95%CI: 57.8% - 65.3%) | 167(25.2%, 95%CI: 21.9%-28.6%) | 664 | | | | |

**Table 5. COVID- 19 contacts tracing, testing, and cluster investigation.**

| Region | Contact tracing and test result | | | | Number of clusters investigated |
|---|---|---|---|---|---|
| | Identified | Tested | % Tested | Positive cases (PR) | |
| Addis Ababa | 3315 | 2147 | 64.8 | 394(18.4%, 95%CI: 16.7%-20.0%)) | 25 |
| Afar | 258 | 229 | 88.8 | 40(17.5%, 95%CI: 12.8%-23.0%) | 4 |
| Amhara | 90 | 56 | 62.2 | 17(30.4%, 95%CI: 18.8%-44.1%) | 20 |
| Benishangul Gumuz | 577 | 505 | 87.5 | 104(20.6%, 95%CI: 15.0%-21.4%) | 56 |
| Dire Dawa | 4584 | 3629 | 79.2 | 242(6.7%, 95%CI: 5.9%-7.5%) | 31 |
| Gambella | 145 | 97 | 66.9 | 12(12.4%, 95%CI: 6.6%-20.6%) | 4 |
| Harari | 564 | 197 | 34.9 | 45(22.8%, 95%CI: 17.2%-29.4%) | 17 |
| Oromia | 1520 | 1517 | 99.8 | 144(9.5%, 95%CI: 8.1%-11.1%) | 0 |
| Sidama | 268 | 42 | 15.7 | 10(23.8%, 95%CI: 12.1%-39.5%) | 0 |
| SNNPR | 895 | 394 | 44.0 | 47(11.9%, 95%CI: 8.9%-15.6%) | 15 |
| Somali | 597 | 510 | 85.4 | 40(7.8%, 95%CI: 5.7%-10.5%) | 32 |
| Grand Total | 12813 | 9323 | 72.8 | 1095(11.8%, 95%CI: 11.1%-12.4%) | 204 |

measures. First, designated surveillance officers in the selected healthcare facilities had the primary responsibility to carry out COVID-19 surveillance activities. Second, monitoring, supportive supervision, and mentorship activities were robust. Third, the political commitment was high, and fourth, electronic DHIS2-based data capturing for COVID-19 was utilized. These implementation lessons need to be scaled up to ensure a sustainable integration of the COVID-19 surveillance system.

Referencing the WHO variant tracking data, we suspect the three waves of COVID-19 outbreaks identified during the one year of surveillance data occurred due to changes in the COVID-19 variants in circulation. The first wave (October 2021) may be attributed to the Delta variant, which was characterized by higher severity (hospitalization), but a relatively lower positivity rate [25,26]. This was when the highest death rate, ICU admission, and mechanical ventilator usage were recorded. Consistent with our findings, other studies have also indicated that the Delta variant has more than 60% increased hospitalization as compared to other variants [25–27]. The second wave (December 2021) may be attributed to the Omicron variant, which was characterized by a higher positivity rate, but lower severity. During this wave, more than one out of three individuals tested were found to be positive for COVID-19. Omicron cases were less symptomatic, resulted in fewer hospital admissions for those who were unvaccinated and for those who were already immunized after the booster dose and were associated with quicker recovery [21,28]. The third wave (January 2022) may be attributed to the Omicron subvariant, which had similar characteristics to the Omicron variant in terms of positivity and severity [21].

The overall positivity rate in our study was relatively lower than study reports from other countries [29–31]. However, there was regional variation in the positivity rate. For instance, consistent with other studies [17,32], nearly one-third of the tested patients in Addis Ababa (the capital city) were found to be positive for COVID-19, while only 1% and 2% positivity were reported in the Afar and Somali regions, respectively. These findings could be due to facilities are close to the point of entry (PoE) in these two regions, where the majority of the tested clients are border-crossing clients. As the population density in Addis Ababa is higher than in the regions, the likelihood of person-to-person contact is higher. Moreover, the lower positivity rates in the regions might be due to the warm weather of the regions, as consistent findings were reported from a study in Pakistan [33].

The COVID-19 CFR among all confirmed cases is higher in our study than the national CFR (2.8 compared to 2.0%) for the same period [1]. This variation could be explained by the

type of surveillance we employed ─ health facility-based versus the national surveillance system, which includes facility-based, community-based, point-of-entry, and mortality surveillance types [10]. It is likely symptomatic and possibly severe COVID-19 cases with comorbidities were captured by facility-based surveillance.

Similar to other countries [34,35], a substantial number of HCWs were found to be infected and died of COVID-19 in Ethiopia. Moreover, Mohammed et al. reported that more than 60% of the HCWs in Ethiopia were hesitant to receive the COVID-19 vaccines in Ethiopia [36]. This has several implications for clinical and public health practices. As the ratio of HCWs to the population in Ethiopia is low, their infection and death substantially affected the capacity of the healthcare system to respond to the pandemic and other diseases. HCWs are part of the basic elements of health systems and their contribution is critical to achieving global health security and universal health coverage [34]. Moreover, the infection and death of HCWs may indicate poor infection prevention and control (IPC) strategies and practices, including limited availability and appropriate use of personal protective equipment (PPE) [37–40]. Therefore, the country must adapt and execute critical precautionary measures to reduce HCW infection and death and ensure a safe and sustainable COVID-19 surveillance system is implemented. Accelerating COVID-19 vaccination among these critical groups, the frontline HCWs, is critical.

This study analyzed data captured from a well-monitored, collaborative COVID-19 surveillance project covering all regions of Ethiopia. The analysis included data captured over a full year period including regional variation in the number (proportion) of tests, cases, deaths, and HCW-associated infections and deaths to fully understand trends and justify identified variations. However, this study has some limitations. Results were not able to capture more fine-scale variation because the study relied on aggregate surveillance data as opposed to individual, patient-level data. COVID-19 surveillance data were not able to be captured in the Tigray region and some parts of the Amhara and Afar regions due to conflict which may limit the generalization of our findings to the country as a whole. Further, the surveillance system may not adequately capture COVID-19 cases with an asymptomatic or mild clinical course at the community level.

## Conclusions

The temporal and regional variations in COVID-19 cases and deaths in the population and among HCWs underscore the need for concerted efforts to address the disparities in the COVID-19 surveillance and response system and the provision of adequate PPE for the HCWs. These lessons should be critically considered during the integration of the COVID-19 surveillance system into the routine surveillance system to monitor its trends, detect variants of concern and other respiratory diseases, and sustain gains made during the acute phase of the COVID-19 response.

## Supporting information

**S1 Checklist.**
(DOCX)

**S1 Appendix. Definition of variables.**
(DOCX)

**S1 Data. Data supporting the findings of the study.**
(CSV)

## Acknowledgments

The authors would like to acknowledge the U.S. CDC, EPHI, RHBs, OSU GOHi, and ICAP at Columbia University for the support provided during the project implementation and manuscript write up. We are also grateful to the surveillance officers deployed in the healthcare facilities and captured the data analyzed in this study.

## Author Contributions

**Conceptualization:** Gizaw Teka, Adane Woldeab, Nebiyu Dereje, Frehywot Eshetu, Leuel Lisanwork, Zelalem Mekuria, Ayalew Aklilu, Jemal Aliy, Sileshi Lulseged, Berhanu Amare, Hulemenaw Delelegn, Getnet Yimer, Ebba Abate, Mesfin Wossen, Zenebe Melaku, Wondwossen Gebreyes, Desmond E. Williams, Aschalew Abayneh.

**Data curation:** Gizaw Teka, Adane Woldeab, Nebiyu Dereje, Frehywot Eshetu, Lehageru Gizachew, Zelalem Tazu, Leuel Lisanwork, Eyasu Tigabu, Ayele Gebeyehu, Adamu Tayachew, Mengistu Biru, Tsegaye Berkessa, Abrham Keraleme, Fentahun Bikale, Wolde Shure, Admikew Agune, Bizuwork Haile, Beza Addis, Muluken Moges, Melaku Gonta, Aster Hailemariam, Laura Binkley, Saira Nawaz, Shu-Hua Wang, Ayalew Aklilu, Jemal Aliy, Abiy Girmay, Abok Patrick, Berhanu Amare, Hulemenaw Delelegn, Sharon Daves, Ebba Abate, Mesfin Wossen.

**Formal analysis:** Gizaw Teka, Adane Woldeab, Nebiyu Dereje, Lehageru Gizachew, Zelalem Tazu, Ayele Gebeyehu.

**Funding acquisition:** Frehywot Eshetu, Bizuwork Haile, Beza Addis, Laura Binkley, Saira Nawaz, Shu-Hua Wang, Zelalem Mekuria, Jemal Aliy, Sileshi Lulseged, Berhanu Amare, Hulemenaw Delelegn, Sharon Daves, Getnet Yimer, Ebba Abate, Mesfin Wossen, Zenebe Melaku, Wondwossen Gebreyes, Desmond E. Williams, Aschalew Abayneh.

**Investigation:** Gizaw Teka, Adane Woldeab, Nebiyu Dereje, Frehywot Eshetu, Lehageru Gizachew, Leuel Lisanwork, Eyasu Tigabu, Mengistu Biru, Abrham Keraleme, Fentahun Bikale, Muluken Moges, Melaku Gonta, Zelalem Mekuria, Jemal Aliy, Getnet Yimer, Ebba Abate, Mesfin Wossen, Zenebe Melaku, Wondwossen Gebreyes, Desmond E. Williams, Aschalew Abayneh.

**Methodology:** Gizaw Teka, Adane Woldeab, Nebiyu Dereje, Frehywot Eshetu, Lehageru Gizachew, Zelalem Tazu, Leuel Lisanwork, Eyasu Tigabu, Ayele Gebeyehu, Adamu Tayachew, Mengistu Biru, Tsegaye Berkessa, Abrham Keraleme, Fentahun Bikale, Wolde Shure, Admikew Agune, Bizuwork Haile, Beza Addis, Muluken Moges, Melaku Gonta, Aster Hailemariam, Laura Binkley, Saira Nawaz, Shu-Hua Wang, Ayalew Aklilu, Jemal Aliy, Sileshi Lulseged, Abiy Girmay, Abok Patrick, Berhanu Amare, Hulemenaw Delelegn, Sharon Daves, Getnet Yimer, Ebba Abate, Mesfin Wossen, Zenebe Melaku, Wondwossen Gebreyes, Desmond E. Williams, Aschalew Abayneh.

**Project administration:** Gizaw Teka, Adane Woldeab, Nebiyu Dereje, Frehywot Eshetu, Lehageru Gizachew, Leuel Lisanwork, Eyasu Tigabu, Adamu Tayachew, Mengistu Biru, Tsegaye Berkessa, Abrham Keraleme, Fentahun Bikale, Wolde Shure, Admikew Agune, Bizuwork Haile, Beza Addis, Muluken Moges, Melaku Gonta, Aster Hailemariam, Saira Nawaz, Zelalem Mekuria, Ayalew Aklilu, Jemal Aliy, Sileshi Lulseged, Berhanu Amare, Sharon Daves, Ebba Abate, Mesfin Wossen, Zenebe Melaku, Wondwossen Gebreyes, Aschalew Abayneh.

**Resources:** Gizaw Teka, Adane Woldeab, Nebiyu Dereje, Leuel Lisanwork, Adamu Tayachew, Tsegaye Berkessa, Bizuwork Haile, Laura Binkley, Zelalem Mekuria, Ayalew Aklilu, Jemal

Aliy, Berhanu Amare, Hulemenaw Delelegn, Ebba Abate, Mesfin Wossen, Zenebe Melaku, Wondwossen Gebreyes, Desmond E. Williams, Aschalew Abayneh.

**Software:** Gizaw Teka, Adane Woldeab, Nebiyu Dereje, Zelalem Tazu, Ayele Gebeyehu, Fentahun Bikale, Ayalew Aklilu.

**Supervision:** Gizaw Teka, Adane Woldeab, Nebiyu Dereje, Frehywot Eshetu, Lehageru Gizachew, Leuel Lisanwork, Eyasu Tigabu, Adamu Tayachew, Mengistu Biru, Tsegaye Berkessa, Abrham Keraleme, Fentahun Bikale, Wolde Shure, Admikew Agune, Bizuwork Haile, Beza Addis, Muluken Moges, Melaku Gonta, Aster Hailemariam, Shu-Hua Wang, Jemal Aliy, Berhanu Amare, Hulemenaw Delelegn, Sharon Daves, Getnet Yimer, Ebba Abate, Mesfin Wossen, Zenebe Melaku, Wondwossen Gebreyes, Desmond E. Williams, Aschalew Abayneh.

**Validation:** Gizaw Teka, Adane Woldeab, Nebiyu Dereje, Lehageru Gizachew, Zelalem Tazu, Eyasu Tigabu, Ayele Gebeyehu, Adamu Tayachew, Mengistu Biru, Tsegaye Berkessa, Abrham Keraleme, Wolde Shure, Admikew Agune, Bizuwork Haile, Beza Addis, Muluken Moges, Melaku Gonta, Aster Hailemariam, Laura Binkley, Saira Nawaz, Shu-Hua Wang, Zelalem Mekuria, Ayalew Aklilu, Jemal Aliy, Sileshi Lulseged, Abiy Girmay, Abok Patrick, Berhanu Amare, Hulemenaw Delelegn, Sharon Daves, Ebba Abate, Mesfin Wossen, Zenebe Melaku, Wondwossen Gebreyes, Desmond E. Williams, Aschalew Abayneh.

**Visualization:** Gizaw Teka, Adane Woldeab, Nebiyu Dereje, Lehageru Gizachew, Zelalem Tazu, Ayele Gebeyehu, Abiy Girmay.

**Writing – original draft:** Gizaw Teka, Adane Woldeab, Nebiyu Dereje, Frehywot Eshetu, Lehageru Gizachew, Zelalem Tazu, Ayele Gebeyehu, Adamu Tayachew, Mengistu Biru, Tsegaye Berkessa, Fentahun Bikale.

**Writing – review & editing:** Gizaw Teka, Adane Woldeab, Nebiyu Dereje, Frehywot Eshetu, Lehageru Gizachew, Zelalem Tazu, Leuel Lisanwork, Eyasu Tigabu, Ayele Gebeyehu, Adamu Tayachew, Mengistu Biru, Tsegaye Berkessa, Abrham Keraleme, Fentahun Bikale, Wolde Shure, Admikew Agune, Bizuwork Haile, Beza Addis, Muluken Moges, Melaku Gonta, Aster Hailemariam, Laura Binkley, Saira Nawaz, Shu-Hua Wang, Zelalem Mekuria, Ayalew Aklilu, Jemal Aliy, Sileshi Lulseged, Abiy Girmay, Abok Patrick, Berhanu Amare, Hulemenaw Delelegn, Sharon Daves, Getnet Yimer, Ebba Abate, Mesfin Wossen, Zenebe Melaku, Wondwossen Gebreyes, Desmond E. Williams, Aschalew Abayneh.

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
