## [Decision Letter · Decision Letter 0]

5 Dec 2023

PGPH-D-23-01272

Regional and temporal variations in COVID-19 cases and deaths in Ethiopia: Lessons learned from the COVID-19 enhanced surveillance and response

Dear Dr. Nebiyu Dereje,

Thank you for submitting your manuscript to PLOS Global Public Health. After careful consideration, we feel that it has merit but does not fully meet PLOS Global Public Health’s publication criteria as it currently stands. Therefore, we invite you to submit a revised version of the manuscript that addresses the points raised during the review process.

We look forward to receiving your revised manuscript.

Kind regards,

Sukanta Chowdhury, Ph.D

Academic Editor

Journal Requirements:

2. We have noticed that you have uploaded Supporting Information files, but you have not included a list of legends. Please add a full list of legends for your Supporting Information files after the references list.

3. In the online submission form, you indicated that "All aggregated COVID-19 surveillance and response data used in this study will be made available in the EPHI data repository after publication. Access to data will be provided following the investigators’ approval of a proposal. Request for the data should be directed to the corresponding author. People requesting data will need to sign a data access agreement that will be forwarded upon request and the database will be transferred electronically". All PLOS journals now require all data underlying the findings described in their manuscript to be freely available to other researchers, either 1. In a public repository, 2. Within the manuscript itself, or 3. Uploaded as supplementary information.

4. Some material included in your submission may be copyrighted. According to PLOS’s copyright policy, authors who use figures or other material (e.g., graphics, clipart, maps) from another author or copyright holder must demonstrate or obtain permission to publish this material under the Creative Commons Attribution 4.0 International (CC BY 4.0) License used by PLOS journals. Please closely review the details of PLOS’s copyright requirements here: PLOS Licenses and Copyright. If you need to request permissions from a copyright holder, you may use PLOS's Copyright Content Permission form.

Potential Copyright Issues:

Figs 2 & 5: please (a) provide a direct link to the base layer of the map (i.e., the country or region border shape) and ensure this is also included in the figure legend; and (b) provide a link to the terms of use / license information for the base layer image or shapefile. We cannot publish proprietary or copyrighted maps (e.g. Google Maps, Mapquest) and the terms of use for your map base layer must be compatible with our CC-BY 4.0 license. 

"

Reviewers' comments:

Reviewer's Responses to Questions

**Comments to the Author**

1. Does this manuscript meet PLOS Global Public Health’s publication criteria? Is the manuscript technically sound, and do the data support the conclusions? The manuscript must describe methodologically and ethically rigorous research with conclusions that are appropriately drawn based on the data presented.

Reviewer #1: Partly

Reviewer #2: Yes

2. Has the statistical analysis been performed appropriately and rigorously?

Reviewer #1: No

Reviewer #2: N/A

3. Have the authors made all data underlying the findings in their manuscript fully available (please refer to the Data Availability Statement at the start of the manuscript PDF file)?

Reviewer #1: Yes

Reviewer #2: Yes

4. Is the manuscript presented in an intelligible fashion and written in standard English?

Reviewer #1: Yes

Reviewer #2: Yes

5. Review Comments to the Author

Reviewer #1: The authors did a good work in delineating the enhanced surveillance of COVID-19 and lessons learned in Ethiopia. Nevertheless, there are some areas in the manuscript need to be strengthened, including the main tables and figures. Please refer to my comments and their corresponding highlighted areas in the attachment.

Regarding statistical analysis, the authors need to consider adding 95% confidence intervals to the tables in main text to indicate uncertainties in the proportion estimates.

Reviewer #2: The paper discusses the regional and temporal variations in COVID-19 cases and deaths in Ethiopia, based on data captured between October 2021 and September 2022. The study analyzed 215,024 individuals' data and found 18,964 COVID-19 cases and 534 deaths. The positivity rate ranged from 1% to 15% across different regions, and three waves of outbreaks were detected during the analysis period. The study also highlighted disparities in the COVID-19 surveillance and response system, emphasizing the need for concerted efforts to address these variations.

1. This paper has worked on data collection categorization and the authors have done a good job in this regard. However, the overall analytical approach of this study is not innovative and simplistic enough and the research conducted is mainly limited to the previous period. I think that research on such a sensitive topic should be more updated in terms of time scales of real-world scenarios to produce some new and meaningful results. I believe that more updated analytical methods could be used to explore some of the latest evolutions and contributions to the recent past.

2. Provide more details on the research methods and data collection process. Explain the time frame for data collection, sample selection methods, and specific steps taken in data analysis to enhance the credibility and reproducibility of the study.

3. Clearly explain the trends in COVID-19 cases and deaths in different regions and periods in the discussion section. Provide more explanation and analysis of these trends, including their underlying causes and potential impacts.

4. Emphasize the importance of addressing disparities in the COVID-19 surveillance and response system highlighted in the study, and provide concrete recommendations for addressing these differences. This could include improving coordination and cooperation among monitoring and response systems in different regions to ensure more equitable and consistent response measures.

5. Discuss the importance of the study's results for COVID-19 response strategies and public health policies. Explore how these findings can be used to guide decision-making and resource allocation for more effective control and management of COVID-19 outbreaks.

6. PLOS authors have the option to publish the peer review history of their article (what does this mean?). If published, this will include your full peer review and any attached files.

**Do you want your identity to be public for this peer review?** For information about this choice, including consent withdrawal, please see our Privacy Policy.

Reviewer #1: **Yes: **Dehao Chen

Reviewer #2: No

---

## [Decision Letter · Decision Letter 1]

1 Apr 2024

PGPH-D-23-01272R1

Regional and temporal variations in COVID-19 cases and deaths in Ethiopia: Lessons learned from the COVID-19 enhanced surveillance and response

Dear Dr. Nebiyu Dereje,,

Thank you for submitting your manuscript to PLOS Global Public Health. After careful consideration, we feel that it has merit but does not fully meet PLOS Global Public Health’s publication criteria as it currently stands. Therefore, we invite you to submit a revised version of the manuscript that addresses the points raised during the review process.

We look forward to receiving your revised manuscript.

Kind regards,

Sukanta Chowdhury, Ph.D

Academic Editor

Journal Requirements:

2. Please send a completed 'Competing Interests' statement, including any COIs declared by your co-authors. If you have no competing interests to declare, please state "The authors have declared that no competing interests exist". Otherwise please declare all competing interests beginning with the statement "I have read the journal's policy and the authors of this manuscript have the following competing interests:"

Additional Editor Comments (if provided):

Table 2: What % of samples tested positive by Ag-RDT and RT-PCR separately? Please include this.

Line 209: More than 200 clusters of outbreaks were identified. I did not find cluster definition in the method section. Suggest to add.

Any death was reported from clusters? Please mention.

Table 5: A large number of cases were detected from clusters. What types of test used to detect cluster cases? Suggest to add information.

Discussion: Add reference for the statement “COVID-19 has had a tremendous impact on the overall health, social, psychological, and economic wellbeing of the Ethiopian population.”

The first wave (October 2021) was may be attributed to the Delta variant. Suggest to include statistics from other study to compare or support your statement.

In the discussion section, authors mentioned three variants name for specific waves but nothing mention about the source of variant information in the method section. Clarify the issue and add information.

Add references for the statement “The overall positivity rate in our study was relatively lower than study reports from other countries.”

Add reference for “The COVID-19 CFR among all confirmed cases is higher in our study than the national CFR (2.8 compared to 2.0%) for the same period.

Any data about the vaccination among HCW? Vaccination impact has been discussed less in the discussion section. Suggest to add information about vaccination in the introduction section.

Reviewers' comments:

Reviewer's Responses to Questions

**Comments to the Author**

1. If the authors have adequately addressed your comments raised in a previous round of review and you feel that this manuscript is now acceptable for publication, you may indicate that here to bypass the “Comments to the Author” section, enter your conflict of interest statement in the “Confidential to Editor” section, and submit your "Accept" recommendation.

Reviewer #1: All comments have been addressed

Reviewer #2: (No Response)

2. Does this manuscript meet PLOS Global Public Health’s publication criteria? Is the manuscript technically sound, and do the data support the conclusions? The manuscript must describe methodologically and ethically rigorous research with conclusions that are appropriately drawn based on the data presented.

Reviewer #1: (No Response)

Reviewer #2: (No Response)

3. Has the statistical analysis been performed appropriately and rigorously?

Reviewer #1: (No Response)

Reviewer #2: (No Response)

4. Have the authors made all data underlying the findings in their manuscript fully available (please refer to the Data Availability Statement at the start of the manuscript PDF file)?

Reviewer #1: (No Response)

Reviewer #2: (No Response)

5. Is the manuscript presented in an intelligible fashion and written in standard English?

Reviewer #1: (No Response)

Reviewer #2: (No Response)

6. Review Comments to the Author

Reviewer #1: (No Response)

Reviewer #2: COVID-19 has come to an end, and the research innovation in this paper is not very obvious, just some simple spatio-temporal pattern analysis. Given that there is relatively little research and reporting on COVID-19 in Africa, I recommend accepting and publishing this paper. I hope that the author team can carefully proofread your paper to ensure that the information is accurate and the text is brief. One more suggestion, please improve the quality of your figures.

7. PLOS authors have the option to publish the peer review history of their article (what does this mean?). If published, this will include your full peer review and any attached files.

**Do you want your identity to be public for this peer review?** For information about this choice, including consent withdrawal, please see our Privacy Policy.

Reviewer #1: No

Reviewer #2: No

---

## [Editor Report · Decision Letter 2]

10 Apr 2024

Regional and temporal variations in COVID-19 cases and deaths in Ethiopia: Lessons learned from the COVID-19 enhanced surveillance and response

PGPH-D-23-01272R2

Dear Dr. Nebiyu Dereje,

We are pleased to inform you that your manuscript 'Regional and temporal variations in COVID-19 cases and deaths in Ethiopia: Lessons learned from the COVID-19 enhanced surveillance and response' has been provisionally accepted for publication in PLOS Global Public Health.

Best regards,

Sukanta Chowdhury, Ph.D

Academic Editor
